# RainMamba: Enhanced Locality Learning with State Space Models for Video Deraining

Hongtao Wu
The Hong Kong University of Science and Technology (Guangzhou)
Guangzhou, China
hwu375@connect.hkust-gz.edu.cn

Yijun Yang
The Hong Kong University of Science and Technology (Guangzhou)
Guangzhou, China
yyang018@connect.hkust-gz.edu.cn

Huihui Xu
The Hong Kong University of Science and Technology (Guangzhou)
Guangzhou, China
hxu047@connect.hkust-gz.edu.cn

Weiming Wang
Hong Kong Metropolitan University
Hong Kong SAR, China
wmwang@hkmu.edu.hk

Jinni Zhou
The Hong Kong University of Science and Technology (Guangzhou)
Guangzhou, China
eejinni@hkust-gz.edu.cn

Lei Zhu*
The Hong Kong University of Science and Technology (Guangzhou) & The Hong Kong University of Science and Technology
leizhu@ust.hk

## ABSTRACT

The outdoor vision systems are frequently contaminated by rain streaks and raindrops, which significantly degenerate the performance of visual tasks and multimedia applications. The nature of videos exhibits redundant temporal cues for rain removal with higher stability. Traditional video deraining methods heavily rely on optical flow estimation and kernel-based manners, which have a limited receptive field. Yet, transformer architectures, while enabling long-term dependencies, bring about a significant increase in computational complexity. Recently, the linear-complexity operator of the state space models (SSMs) has contrarily facilitated efficient long-term temporal modeling, which is crucial for rain streaks and raindrops removal in videos. Unexpectedly, its unidimensional sequential process on videos destroys the local correlations across the spatio-temporal dimension by distancing adjacent pixels. To address this, we present an improved SSMs-based video deraining network (RainMamba) with a novel Hilbert scanning mechanism to better capture sequence-level local information. We also introduce a difference-guided dynamic contrastive locality learning strategy to enhance the patch-level self-similarity learning ability of the proposed network. Extensive experiments on four synthesized video deraining datasets and real-world rainy videos demonstrate the superiority of our network in the removal of rain streaks and raindrops. Our code and results are available at https://github.com/TonyHongtaoWu/RainMamba.

---

*Corresponding author.

---

## CCS CONCEPTS

• **Computing methodologies** → **Reconstruction**; **Computer vision tasks**.

## KEYWORDS

Video deraining, State space models, Hilbert scan

## 1 INTRODUCTION

Videos captured from outdoor systems, *i.e.*, surveillance cameras and mobile sensors in autonomous vehicles, are often corrupted by both rain streaks and raindrops. These degradations significantly damage the visual perceptual quality and tend to degenerate the performance of subsequent outdoor computer vision and multimedia computing tasks, *e.g.*, object detection [1, 14], semantic segmentation [55, 57] and autonomous driving [17, 89]. Therefore, rain removal is an indispensable pre-processing step to enhance the robustness and stability of outdoor intelligent systems and multimedia applications.

The earliest video deraining methods [5, 19, 50, 81] were designed based on handcraft priors and attempted to model the physical characteristics and photometric properties of the rain layer to solve the problem. However, their performance is severely restricted by the laborious priors and they encounter complex optimization challenges. In recent years, deep learning-based methods, including convolutional neural networks (CNN) and recurrent neural networks (RNN), have demonstrated effectiveness in video deraining tasks. The success of video deraining methods [40, 68, 71–74, 83] hinges on some elaborated modules, such as optical flow estimation and deformable convolutions, to leveraging temporal corrections from neighboring frames. Nevertheless, these CNN-based methods usually have limited spatial-temporal receptive fields compared to recent transformer-based architectures [36, 56, 75, 79, 82]. On the other hand, while these transformer-based methods achieve global understanding on videos, they heavily rely on Multi-Head Self-attention (MSA) mechanism, resulting in quadratic complexity related to sequence length. This poses significant efficiency challenges in handling long video sequences [2] and impedes its application in modeling long-term inter-frame correlation.

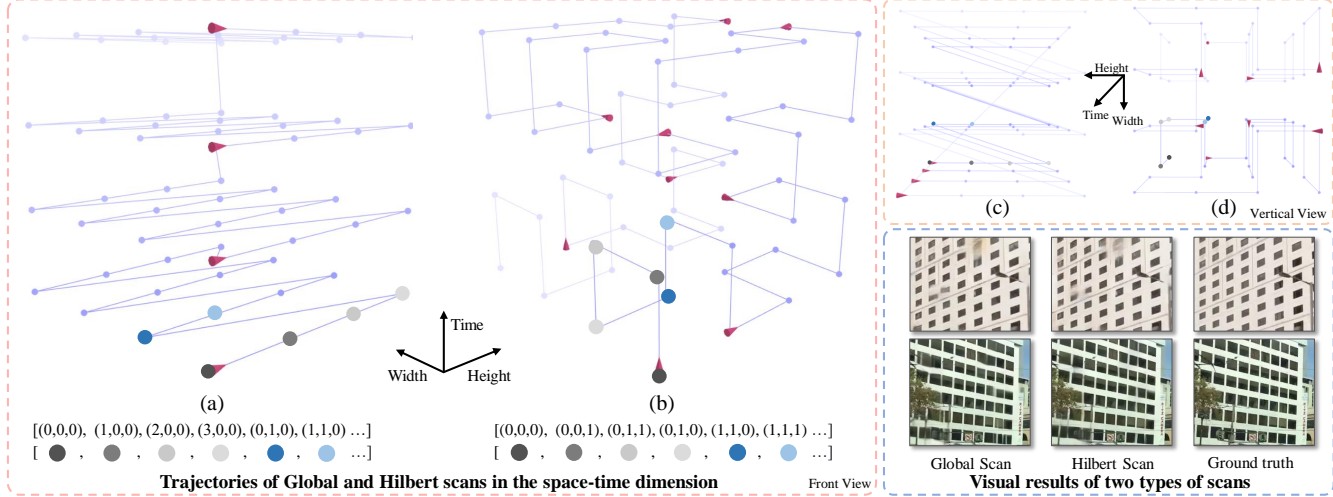

Figure 1: Motivation illustration and visual comparisons of two different scanning methods. (a) and (c) are the illustration of the global scan method, while (b) and (d) are the illustration of the Hilbert scan method. The temporal scanning differences are emphasized in (a) and (b), whereas the spatial scanning differences are depicted in (c) and (d). The lines and endpoints (represent pixel points) are shaded in gradients from dark to light, signifying the path of the scan. For a more intuitive understanding, please refer to the dynamic display in the *Supplementary Video*. We leverage the Hilbert curve's locality feature to improve the utilization of local information in the time-space dimension during the scanning process. The visual results indicate that our local scanning mechanism improves spatial structure preservation of derived results.

Recently, State Space Models (SSMs) [32], deriving from control systems theory, have demonstrated their advantages in natural language processing (NLP) [20, 21] and computer vision [41, 90] by their linear complexity in handling long sequences. By formalizing discrete state space equations recursively, Mamba can capture long-range dependencies [20], thereby improving video reconstruction quality by establishing global connections among pixels. Nonetheless, the conventional Mamba model [20], tailored for 1D sequential data in NLP, encounters inherent bottlenecks in video restoration tasks due to local pixel forgetting. As illustrated in Fig. 1(a), the Mamba model recursively processes frames flattened into 1D sequences. This approach unexpectedly destroys the causality between spatio-temporally adjacent pixels in the video sequences, leading to the absence of local information in sequence-level temporal modeling.

To address the aforementioned issues, we devise an improved SSMs-based framework dubbed RainMamba for rain removal in videos. We present a novel Hilbert scan mechanism that leverages the inherent locality characteristic [3, 44] of Hilbert curve [26] to enhance the locality learning of the vanilla Mamba. In particular, we convert the spatio-temporal pixels into a 1D sequence following the trajectory of the Hilbert curve, thus leveraging the Hilbert curve's locality preservation for fine-grained restoration. However, flattening video data into one 1D sequence inevitably diminishes the inherent local spatial correlations, hindering the restoration of rain streaks and raindrops areas. Based on the observation that the patches in a given frame exhibit similarity with the neighboring counterpart in the same frame and subsequent frames, we propose a difference-guided dynamic contrastive locality learning strategy to preserve patch-level semantics. Specifically, we derive rain location from the difference map for anchor sampling, while selecting the

spatio-temporally surrounding patches as positive samples and spatio-temporally distant patches with significant differences as negative samples. Motivated by Curriculum Learning [7], we also introduce a dynamic mechanism that facilitates the optimization by adjusting the sampling distance for positive and negative samples.

In summary, our contributions are as follows:

- We propose the first framework to adapt state space models to video deraining tasks by the Coarse-to-Fine Mamba Block.
- We equip SSMs with a novel Hilbert scanning mechanism, which achieves localized scanning across both temporal and spatial dimensions. This approach substantially improves our models' ability to explore sequence-level local information.
- We introduce a difference-guided dynamic contrastive locality learning approach to enhance the patch-level self-similarity learning ability.
- Experimental results on four synthesized video deraining datasets and real-world rainy videos demonstrate that our network prominently outperforms state-of-the-art methods.

## 2 RELATED WORK

### 2.1 Video Deraining

In the past decades, deep learning-based methods [15, 18, 38, 48] have achieved impressive results for rain streak and raindrop removal in images. SPANet [60] introduced the four-directional Initialize Recurrent Neural Networks to obtain the contextual information for rain streaks removal. CCN [47] employed neural architecture search to adaptively find an optimal architecture for jointly tackling rain streaks and raindrops removal. D-DAiAM [86] leveraged the output differences between multiple deraining stages consisting of dual attention-in-attention model for joint removal

of rain streaks and raindrops. Recently, Video-level deraining techniques [49, 54, 59, 71, 72, 74, 77, 83, 87] have been developed to leverage temporal correlation and information across sequential frames to improve deraining results. FCRVD [72] constructed a two-stage recurrent network incorporating dual flow constraints to enhance motion information for video rain streaks and accumulation flow removal. SAVD [71] employed deformable convolution to align multi-frame features, effectively removing both rain streaks and rain accumulation in videos. RDD-Net [59] introduced the rain streak motion prior to a recurrent video rain streaks removal network. ESTINet [87] employed long-short term memory to effectively capture spatio-temporal features and temporal correlations between neighboring frames. Considering the distinct physical properties of raindrops, VWR [65] developed a spatio-temporal attention mechanism for video raindrops removal. As rain streaks and raindrops frequently co-occur in videos, Wu *et al.* [68] devised a video deraining network ViMP-Net that integrates optical flow and mask-guided intra-frame and inter-frame transformers to sequentially remove rain streaks and raindrops. In contrast, we introduce SSMs to causally model temporal information, replacing the mechanisms with low efficiency based on optical flow, deformable kernel or self-attention.

## 2.2 State Space Models

Recently, State Space Models (SSMs) [32] have demonstrated notable efficiency in utilizing state space transformations [22] to manage long-term dependencies within language sequences. S4 [21] introduced a structured state-space sequence model to exploit long-range dependencies with the benefit of linear complexity. Based on this, Mamba [20] integrates efficient hardware design and a selection mechanism employing parallel scan (S6), thereby surpassing Transformers in processing extensive natural language sequences. Subsequently, S4ND [45] explores SSMs' continuous-signal modeling of multi-dimensional data like images and videos. More recently, Vision Mamba [90] and Vmamba [41] pioneered generic vision tasks, introducing bi-directional scan and cross-scan mechanisms to tackle the directional sensitivity challenge in SSMs. Thanks to the superiority in addressing complex vision challenges, SSMs have been integrated into many vision tasks including object detection [28], image segmentation [69], image restoration [23, 52], video object segmentation [70, 78] and video understanding [34]. To the best of our knowledge, SSMs have not yet been explored in the video deraining task. Unlike previous SSMs-based works, we introduce a Hilbert scanning approach to enhance its sequence-level locality learning in spatio-temporal modeling.

## 2.3 Contrastive Learning

Contrastive learning, an effective self-supervised learning technique [12, 25, 46, 76], seeks to minimize the distance between anchors and positive samples while maximizing the distance from negative samples within the representation space. Contrastive learning has been explored in some low-level vision tasks like image translation [24], image super-resolution [63], image dehazing [66], unsupervised image deraining [16, 80], image-to-image translation [24], video desnowing [10, 67] and video deraining [58] . DCD-GAN [16] utilized contrastive learning to leverage features

from unpaired clean images for guiding rain removal in the latent. ANLCL [80] formulated an unsupervised contrastive learning based on the self-similarity property within samples and the mutually exclusive property between the rain layer and image layer. UVDEC [58] constructed a cross-modal contrastive learning to investigate the mutual exclusion and similarity of rain-background layers between pixel domain and event domain. The pivotal aspect of contrastive learning depends on the selection of anchor, positive, and negative samples. Unlike previous image-level methods that sampled based on input and predictions or randomly selected patches, we developed a dynamic temporal sampling method to explore spatio-temporal self-similarity between video frames.

## 3 METHODS

### 3.1 Network Architecture

Figure 2 displays the overview of the proposed RainMamba for video deraining task. Specifically, our video deraining network is composed of a feature encoder, a series of cascaded Coarse-to-Fine Mamba modules, and a feature decoder. Given a rainy video sequence $\{I_t \in \mathbb{R}^{3 \times H \times W} \mid t \in [0, T)\}$, we employ a universal backbone (i.e., ConvNeXt [42]) and a lightweight head as the encoder to extract the shallow feature $\{E_t \in \mathbb{R}^{C \times H/4 \times W/4} \mid t \in [0, T)\}$, where $H$ and $W$ denote the height and width of the input frames, and $C$ signifies the number of channels. To accommodate various patterns of rain in frames, we utilize convolutional layers to downsample and upsample the extracted features, thereby learning background semantics in a multi-scale manner. In such a way, the feature is successively fed into Coarse-to-Fine Mamba Module (CFM) to causally learn the global and local temporal correlation and alignment by two different submodules at different scales, *i.e.*, global mamba block (GMB, see 3.3.1) and local mamba block (LMB, see 3.3.2). Subsequently, after multi-scale globality and locality learning from temporal feature, we add $N_3$ CFM modules for further feature refinement. Finally, the enhanced feature by CFM is processed through a reconstruction decoder, which is composed of multiple 3D convolutions and upsample layers, to generate the recovered frames $\{J_t \in \mathbb{R}^{3 \times H \times W} \mid t \in [0, T)\}$. During training, we design and enforce the difference-guided dynamic contrastive regularization to promote the network's dynamic learning of patch-level local self-similarity.

### 3.2 Preliminaries

*3.2.1 State Space Models.* Inspired by continuous linear time-invariant systems, Structured State Space Models (S4) and Mamba represent a class of sequence models that map a one-dimensional sequence $x(k) \in \mathbb{R} \rightarrow y(k) \in \mathbb{R}$ through an intermediate hidden state $h(k) \in \mathbb{R}^{N \times 1}$, where $N$ is the hidden state size. Mathematically, SSMs utilize the ordinary differential equation (ODE) below to transform the input data:

$$\begin{aligned} h'(k) &= \mathbf{A}h(k) + \mathbf{B}x(k), \\ y(k) &= \mathbf{C}h(k), \end{aligned} \tag{1}$$

where $\mathbf{A} \in \mathbb{R}^{N \times N}$ represents the system's evolution parameter, and $\mathbf{B} \in \mathbb{R}^{N \times 1}$, $\mathbf{C} \in \mathbb{R}^{1 \times N}$ are the projection parameters, respectively. For practical application in deep learning, the continuous system described by Eq. 1 is transformed into its discrete counterparts,

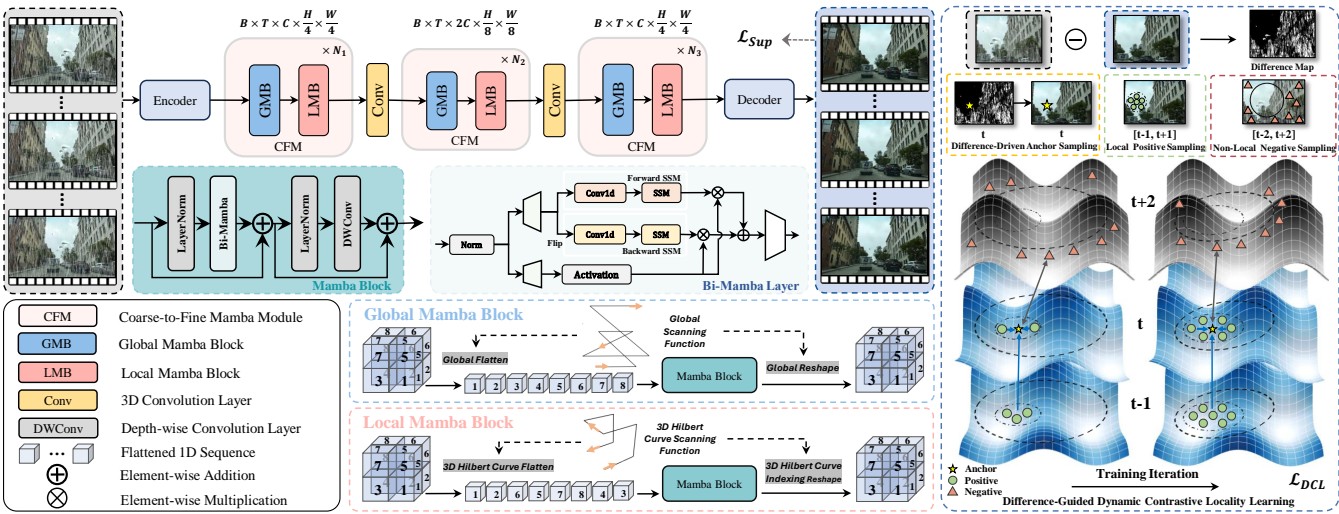

**Figure 2: The architecture of our proposed framework RainMamba for video deraining task. Given a sequence of rainy video frames, the cascading Coarse-to-Fine Mamba Module (CFM) receives the encoded features as input and causally models temporal corrections by the improved state space models (SSMs). The CFM employs Global Mamba Block (GMB) and Local Mamba Block (LMB) to capture sequence-level global and local spatio-temporal dependencies. We develop a novel Hilbert scanning paradigm in LMB to promote the Mamba's locality learning. Moreover, we construct a difference-guided dynamic contrastive locality learning approach to enhance patch-level locality learning. Specifically, we utilize the difference between the input and the ground truth to select the anchor, sampling the positive patch at a spatio-temporally adjacent location to the anchor, and the negative patch at a more distant location. As training progresses, the sampling space for positive samples expands while that for negative samples contracts.**

through a discretization process using the zero-order hold (ZOH) method. This involves a timescale parameter $\Delta \in \mathbb{R} > 0$, converting continuous parameters $(\mathbf{A}, \mathbf{B})$ into discrete parameters $(\overline{\mathbf{A}}, \overline{\mathbf{B}})$, which can be defined as follows:

$$
\begin{aligned}
\overline{\mathbf{A}} &= \exp(\Delta \mathbf{A}), \\
\overline{\mathbf{B}} &= (\Delta \mathbf{A})^{-1}(\exp(\Delta \mathbf{A}) - \mathbf{I}) \cdot \Delta \mathbf{B}.
\end{aligned}
\tag{2}
$$

This results in the following discretized model formulation:

$$
h_k = \overline{\mathbf{A}} h_{k-1} + \overline{\mathbf{B}} x_k, \quad y_k = \mathbf{C} h_k.
\tag{3}
$$

To enhance computational efficiency and scalability, the Eq. 3 can be mathematically transformed into an equivalent CNN form, leveraging parallel computation via a global convolution operation:

$$
\begin{aligned}
\overline{\mathbf{M}} &= \left( \mathbf{C}\overline{\mathbf{B}}, \mathbf{C}\overline{\mathbf{A}}\overline{\mathbf{B}}, \dots, \mathbf{C}\overline{\mathbf{A}}^{L-1}\overline{\mathbf{B}} \right), \\
y &= x \circledast \overline{\mathbf{M}},
\end{aligned}
\tag{4}
$$

where $\overline{\mathbf{M}} \in \mathbb{R}^L$ is a convolutional kernel of the SSM, $L$ is the length of the input sequence x and $\circledast$ represents the convolution operation.

Unlike traditional SSMs that employ constant transition parameters $(\overline{\mathbf{A}}, \overline{\mathbf{B}})$, S6 [20] establish an input-dependent mechanism for matrices $B, C$ and $\Delta$, which enables better perception of input context information and dynamic updates of these parameters.

*3.2.2 Hilbert Curve.* Hilbert curve [26] is a space filling curve (SFC) [43], extensively deployed in various fields, including database [4] and image compression [37]. As illustrated in Figure 1, the Hilbert curve has the ability to connect all elements within a space, and is often utilized as a fractal function [51]. The Hilbert curve's defining characteristic lies in its strong capability to *preserve*

*locality* [29] when transforming from one-dimensional to multi-dimensional spaces, significantly improving feature clustering [13]. Moreover, [44] shown that the Hilbert curve achieves better clustering in three-dimensional spaces. Based on this, Hilbert scan can enhance the correlation of spatio-temporally adjacent features by promoting clustering of neighboring tokens on the sequence. More formally, a SFC can be denoted as $p : [0, 1] \rightarrow [0, 1] \times [0, 1]$, which maps any point from one-dimensional interval $[0, 1]$ to a coordinate in two-dimensional unit square. We also denote $n$ as the curve order of the Hilbert curve, which, in our discrete case, approximates to height and width of each frame. For any two points $u, v$ in $[0,1]$, their space to linear ratio (SLR) is defined as:

$$
SLR = \frac{|\sigma(u) - \sigma(v)|^2}{|u - v|}.
\tag{5}
$$

The dilation factor (DF) of a SFC is defined as the upper bound of the SLR. For the same two points in $[0, 1]$, if an SFC has lower DF, their mappings will also be closer in the unit square, which accords with the locality preserving requirement in the scanning stage. As proved in [6, 13], the DF of Hilbert curve is 6, while the normal Zigzag (row-and-column-order) curve is $4^n - 2^{n+1} + 2$, which diverges to $\infty$ as the curve order $n$ increases. Therefore, as the image resolution increases, the Hilbert curve can better maintain the locality of mapping any two points on a one-dimensional sequence to a multi-dimensional space than the Zigzag curve. Also, the DF can provide explicit mathematical interpretation for comparing the impact of two scans. Specifically, compared to Hilbert scan's locality, Zigzag scan can capture more global correlations. Our research extends the locality of Hilbert curves to video sequences for enhancing the preservation of local pixel information during spatial-temporal

scanning of SSMs. In this manner, the two scanning mechanisms can complement each other in establishing both global and local correlations.

## 3.3 Coarse-to-Fine Mamba Module

Previous works [68, 71, 72, 74] tend to elaborate some modules based on optical flow, deformable kernel or quadratic-complexity self-attention, to exploit temporal information with low efficiency for rain removal in videos. Our method introduces the Structured State Space Models (dubbed Mamba), which can employ the selective scan mechanism to causally process the temporal data with linear complexity. Specifically, we flatten the 3D video data into one-dimensional sequences in two different ways to effectively leverage spatio-temporal corrections from neighboring frames. Our proposed Coarse-to-Fine Mamba Module consisting of Global Mamba Block and Local Mamba Block can progressively mitigate the degradation through holistic and regional multi-scale perception.

*3.3.1 Global Mamba Block.* As illustrated in Figure 2, in Global Mamba Block, we apply a zigzag order approach (i.e. row by row) to construct global flattening. This global scanning mechanism enables SSMs to establish global contextual dependencies between each pixel and all other pixels in a linearized sequence. We transform the input video feature $E \in \mathbb{R}^{C \times T \times H/4 \times W/4}$ into a one-dimensional long sequence $V \in \mathbb{R}^{C \times (T \times H/4 \times W/4)}$ in a spatially-prioritized manner. Then, the flattened sequence $V$ is fed into a Mamba Block. We incorporate the Bi-Mamba layer [90] into our Mamba block for video deraining. This Bi-Mamba layer processes flattened visual sequences through a bidirectional fashion, *i.e.*, forward and backward SSMs, which has been proven to be effective on low-level video tasks [8, 36, 87]. Subsequently, a depth-wise convolution layer with the kernel size of $3 \times 3 \times 3$ is employed to preserve fine-grained details. The operation within the stacked Mamba Block can be defined as follows:

$$V^l = \mathrm{BM}\left(\mathrm{LN}\left(V^{l-1}\right)\right) + V^{l-1}, V^l = \mathrm{DWC}\left(\mathrm{LN}\left(V^l\right)\right) + V^l. \quad (6)$$

Finally, we reshape the output one-dimensional sequence features back to their original three-dimensional features $\hat{E} \in \mathbb{R}^{C \times T \times H/4 \times W/4}$ in the global flattened order.

*3.3.2 Local Mamba Block.* While global scanning facilitates modeling temporal information by causally processing time series data with SSMs, it destroys the inherent local correlations in videos. As shown in Fig. 1(a) and (c), the global scanning approach significantly increases the distance between spatially and temporally adjacent pixels, which leads to severe adjacent pixel forgetting. To overcome this challenge, we present a novel Hilbert scanning technique to enhance the locality learning of video data during scanning. The proposed strategy is illustrated in Fig.1(b) and (d), highlighting the contrast between our method and global scanning approach. As shown in Fig. 2, we first construct a 3D Hilbert curve that traverses every point following the principles of Hilbert curves. In essence, the 3D Hilbert algorithm is designed to recursively subdivide the cubic space and generate Hilbert curve segments within each smaller cubic space. These smaller segments are then merged to construct the full 3D Hilbert curve. Subsequently, we transform each coordinate $(\hat{x}, \hat{y}, \hat{z})$ on the 3D Hilbert curve into a unique one-dimensional index, thereby flattening the three-dimensional space by the Hilbert curve's scanning trajectory. This indexing formula

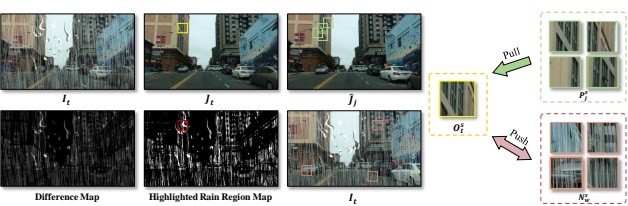

**Figure 3: The motivation and operation of our proposed Difference-Guided Dynamic Contrastive Locality Learning.**

can be represented as:

$$\mathrm{Index} = \hat{x} \cdot Q \cdot T + \hat{y} \cdot T + \hat{z}, \quad (7)$$

where $Q$ is the width length of input features. Based on the Hilbert index, we flatten the input features $\hat{E} \in \mathbb{R}^{C \times T \times H/4 \times W/4}$ into the one-dimensional sequence $\hat{V} \in \mathbb{R}^{C \times (H/4 \times W/4 \times T)}$ which is locally enhanced against the global sequence $V$. Afterward, the sequence is fed into the Mamba block, which has the same network architecture as the global counterpart, to exploit spatio-temporal corrections in the local spirit. Finally, we reshape the output sequence back to its original shape to construct the locally-enhanced features $\hat{E}' \in \mathbb{R}^{C \times T \times H/4 \times W/4}$. In contrast to the global scanning approach, our Hilbert scanning method is designed to more effectively capture sequence-level local dependencies across both temporal and spatial dimensions. This benefits from the Hilbert curve's inherent capacity to preserve locality, as highlighted in Sec. 3.2.2. By cascading Global Mamba Block and Local Mamba Block, our network first obtains the holistic understanding of spatio-temporal information and then preserves its local details following the spirit of Coarse-to-Fine.

## 3.4 Difference-Guided Dynamic Contrastive Locality Learning

Although Mamba can effectively take advantage of the causality of videos by flattening 1D causal sequential data for temporal modeling, it frequently underestimates inherent 2D spatial relationships. To address this, we introduce a regularization strategy to enhance the model's intra- and inter-frame perception of spatial correlations. Inspired by non-local prior [39, 62], we encourage the locality learning of spatial-temporally neighboring pixels by constructing a contrast learning mechanism that exploits the self-similarity from the patch level. The rain in videos exhibits varying intensities and types, making rain removal more challenging especially when rain streaks and raindrops exist simultaneously. According to [47, 68], the $t$-th video frame containing rain streaks and raindrops can be represented as:

$$RSD_t = (1 - M_t) \odot (B_t + S_t) + M_t \odot D_t, \quad (8)$$

where $RSD_t$ denotes the rainy video frame containing the clean background layer $B_t$, rain streaks layer $S_t$ and raindrops layer $D_t$. $M_t$ is a binary mask, and $M_t(x) = 1$ means pixel $x$ is a part of the region occluded by raindrops. Based on Eq. 8, we can derive the position information of rain streaks and raindrops by the difference map $\Omega_t$ between rainy and clean frames.

$$\Omega_t = (1 - M_t) \odot S_t + M_t \odot D_t - M_t \odot B_t. \quad (9)$$

As illustrated in Fig. 3, it's observed that regions containing streaks and raindrops typically correspond to high-response areas on the difference map. We adopt the average threshold method to

select patches with high response in the difference map from the restored frame $\{J_t \mid t \in [0, T)\}$ as anchors $\{O_t \mid t \in [0, T)\}$. We focus the attention of recovery on heavily degraded areas by eliminating the patches with lower response than the average response.

Moreover, we can observe that patches of the target frame share similarities with their adjacent counterparts in the same frame and also subsequent frames. Conversely, these patches are significantly different from the spatio-temporally distant patches. Based on these observations, we select spatially adjacent patches from the neighboring clean frames $\{\hat{J}_j \mid j \in [t-1, t+1]\}$ as positive samples $\{P_j \mid j \in [t-1, t+1]\}$. On the other hand, we treat spatially distant patches from the entire degraded input sequence as negative samples $\{N_w \mid w \in [0, T)\}$ and enhance their distinctiveness from positive samples by performing data augmentation such as rotation, flipping, and blurring. Furthermore, we introduce dynamic learning to facilitate the optimization of intra- and inter-frame locality learning. We incrementally increase the spatial distance $p$ between positive samples and the anchor, and simultaneously decrease the spatial distance $d$ between negative samples and the anchor during the training process. The entire dynamic learning process can be represented as:

$$d = \max(d_0 \cdot \theta^{\frac{e}{m}}, d_{min}), \tag{10}$$

$$p = \min\left(p_0 + \frac{e}{m} \cdot (p_{max} - p_0), p_{max}\right), \tag{11}$$

where $d_0$ is the initial minimum negative distance, $\theta$ denotes the decay rate, $e$ is the number of completed training steps, $m$ is the total number of training steps, $p_0$ indicates the initial positive range and $p$ is the sampling distance of positive samples. This dynamic learning approach enables the network to progressively master patch-level details, advancing from simple to complex concepts.

Ultimately, we employ contrastive learning to ensure the $s - th$ sample $O_t^s$ is pulled closer to positive samples $P_j^s$ and pushed far away from the strongly augmented degraded negative samples $N_w^s$ through the pre-trained VGG feature extractor. Our proposed contrastive learning loss can be formulated as:

$$\mathcal{L}_{DCL} = \frac{1}{S} \sum_{s=1}^{S} \left( \sum_{r=1}^{2} \frac{\mathcal{L}_{L_1}\left(G_r(P_j^s), G_r(O_t^s)\right)}{\mathcal{L}_{L_1}\left(G_r(N_w^s), G_r(O_t^s)\right)} \right), \tag{12}$$

where $\{G_r \mid r \in [1, 2]\}$ extracts the $r - th$ low-level hidden layer features from the pre-trained VGG-19 [53] model.

## 3.5 Loss Function

We adopt the Charbonnier loss [9] and the perceptual loss [31] to improve the visual quality of the restored results. The perceptual loss is to quantify the discrepancy between the features of the prediction and the ground truth. The overall supervised loss is formulated as:

$$\mathcal{L}_{sup} = \mathcal{L}_{pixel} + \lambda_1 \mathcal{L}_{perceptual} + \lambda_2 \mathcal{L}_{DCL}, \tag{13}$$

where $\lambda_1$ and $\lambda_2$ are the balancing hyper-parameters, empirically set as 0.3 and 0.1, respectively. And the perceptual loss is formulated as follows:

$$\mathcal{L}_{perceptual} = \mathcal{L}_{MSE}\left(VGG_{3,8,15}(\hat{J}_t), VGG_{3,8,15}(J_t)\right). \tag{14}$$

## 4 EXPERIMENTS

### 4.1 Dataset and Metric

In this section, we conduct a comparative evaluation of our video deraining network against state-of-the-art methods on four benchmark datasets, including two video rain streak removal datasets, a video raindrop removal dataset, and a video rain streak and raindrop removal dataset. We utilize the peak signal-to-noise ratio (PSNR), the structural similarity index (SSIM) [64], and the learned perceptual image patch similarity (LPIPS) [88] to quantitatively compare different methods.

**Video Rain Streak Removal Datasets.** These two video rain streak removal datasets are RainVID&SS (Rain Video Detection and Semantic Segmentation)[54] and RainSynAll100 [73]. RainVID&SS includes 205 short clips from ImageNet-VID and 3 long clips from CamVid for training. The training set has 86 short clips and 2 long clips. The RainSynAll100 dataset comprises 900 videos for training and 100 videos for testing.

**Video Raindrop Removal Dataset.** LWDDS (Large-scale Waterdrop Dataset for Driving Scenes) [65] is the first synthetic video waterdrop dataset for raindrop removal in driving scenes. This dataset has 67,500 triplets from 45 videos for training, and 600 triplets obtained from 6 videos for testing.

**Video Rain Streak and Raindrop Removal Dataset.** VRDS [68] is the only video raindrop and rain streak removal dataset with a total of 102 videos. 72 videos with 7200 frames are used for training, while 30 videos with 3000 frames are for testing.

### 4.2 Implementation Details.

Our network is trained on NVIDIA RTX 4090 GPUs and implemented on the Pytorch platform. Note that we have four benchmark datsets for testing our network and compared methods. Due to the page limit, we here provide the training details of the VRDS dataset. Specifically, at each training iteration, the input frame is randomly cropped to a spatial resolution of 256×256, and the number of frames per video clip is 5. The total number of the training iteration is 300K. We adopt the Adam optimizer [33] and the polynomial scheduler with a power of 1.0. The initial learning rate of our network is set to $5 \times 10^{-4}$ with a batch size of 8 and a warm-up start of 2k iterations. Moreover, the encoder adopts the ImageNet pre-trained ConvNeXt [42] backbone. The number of Coarse to Fine Mamba Modules $N_1$, $N_2$, and $N_3$ are set to 2, 3, and 2 respectively. For training details on the other three datasets, please refer to *Supplementary Material* or our arXiv version.

### 4.3 Comparisons with State-of-the-art Methods

*4.3.1 Quantitative Comparisons on the VRDS Dataset.* As shown in Tab. 1, we quantitatively compare our proposed method with 12 state-of-the-art (SOTA) image and video deraining methods, including CCN [47], PreNet [48], DRSformer [15], MPRNet [85], Restormer [84], S2VD [83], SLDNet [74], ESTINet [87], RDD [59], RVRT [36], BasicVSR++ [8] and ViMP-Net [68]. Among the 12 compared methods, ViMP-Net has the best PSNR, SSIM, and LPIPS performance, and the PSNR, SSIM, and LPIPS scores are 31.02 dB, 0.9283, and 0.0862. Moreover, our method has better metric results than ViMP-Net, and our PSNR, SSIM, and LPIPS scores are 32.04 dB, 0.9366, and 0.0684. It indicates that the state space model with enhanced locality learning enables our framework to achieve a

**Table 1: Quantitative comparisons between our network and SOTA methods on the VRDS dataset [68]. Results of compared methods are from ViMP-Net [68].**

| Methods | CCN[47] | PreNet[48] | DRSformer[15] | MPRNet[85] | Restormer[84] | S2VD[83] | SLDNet[74] | ESTINet[87] | RDD[59] | RVRT[36] | BasicVSR++[8] | ViMP-Net [68] | Ours |
|---|---|---|---|---|---|---|---|---|---|---|---|---|---|
| PSNR↑ | 23.75 | 27.13 | 28.54 | 29.53 | 29.59 | 18.95 | 23.65 | 27.17 | 28.39 | 28.24 | 29.75 | 31.02 | **32.04** |
| SSIM↑ | 0.8410 | 0.9014 | 0.9075 | 0.9175 | 0.9206 | 0.6630 | 0.8736 | 0.8436 | 0.9096 | 0.8857 | 0.9171 | 0.9283 | **0.9366** |
| LPIPS↓ | 0.2091 | 0.1266 | 0.1143 | 0.0987 | 0.0925 | 0.2833 | 0.1790 | 0.2253 | 0.1168 | 0.1438 | 0.1023 | 0.0862 | **0.0684** |

**Table 2: Quantitative comparison between our network SOTA video deraining methods on the two datasets of RainVID&SS dataset [54]. Results of compared methods are from MPEVNet [54].**

| Datasets | ImageNet-VID+ | | | | | | Cam-Vid+ | | | | | | |
|---|---|---|---|---|---|---|---|---|---|---|---|---|---|
| Methods | MSCSC [35] | FastDrain [30] | PReNet[48] | S2VD [83] | MPEVNet [54] | Ours | MSCSC [35] | FastDrain [30] | PReNet[48] | SLDNet[74] | S2VD [83] | MPEVNet [54] | Ours |
| PSNR↑ | 18.41 | 17.08 | 24.73 | 29.92 | 33.83 | **35.07** | 21.22 | 19.94 | 25.33 | 18.97 | 29.11 | 32.55 | **32.65** |
| SSIM↑ | 0.5148 | 0.4381 | 0.7393 | 0.9228 | 0.9452 | **0.9561** | 0.5515 | 0.4830 | 0.7647 | 0.6267 | 0.8899 | 0.9234 | **0.9328** |

**Table 3: Quantitative comparisons between our network and SOTA video deraining methods on the RainSynAll100 dataset [73]. Note that the PSNR and SSIM results of compared methods are from SALN [79] and NCFL [27].**

| Methods | FastDerain [30] | FCRVD [72] | RMFD [73] | BasicVSR++[8] | NCFL [27] | SALN [79] | Ours |
|---|---|---|---|---|---|---|---|
| PSNR | 17.09 | 21.06 | 25.14 | 27.67 | 28.11 | 29.78 | **32.16** |
| SSIM | 0.5824 | 0.7405 | 0.9172 | 0.9135 | 0.9235 | 0.9315 | **0.9446** |

**Table 4: Quantitative comparisons between our method and SOTA video deraining methods on the LWDDS dataset [65]. The results of compared methods are from SALN [79].**

| Methods | CCN[47] | Vid2Vid [61] | VWR[65] | BasicVSR++[8] | ViMP-Net [68] | SALN [79] | Ours |
|---|---|---|---|---|---|---|---|
| PSNR↑ | 27.53 | 28.73 | 30.72 | 32.37 | 34.22 | 36.57 | **37.21** |
| SSIM↑ | 0.922 | 0.9542 | 0.9726 | 0.9792 | 0.9784 | 0.9802 | **0.9816** |

**Table 5: Quantitative results of our network and constructed baseline networks of the ablation study on the VRDS dataset.**

| Model | GMB | LMB | DCL | PSNR↑ | SSIM↑ | LPIPS↓ | GFLOPs | Parameters(M) | Inference time(s) | Runtime(s/frame) |
|---|---|---|---|---|---|---|---|---|---|---|
| M1 | | | | 29.35 | 0.9118 | 0.0893 | 49.82 | 30.23 | 0.0190 | 0.0038 |
| M2 | ✓ | | | 30.86 | 0.9277 | 0.0774 | 84.40 | 32.49 | 0.0348 | 0.0070 |
| M3 | | ✓ | | 31.04 | 0.9302 | 0.0753 | 84.40 | 32.49 | 0.0355 | 0.0071 |
| M4 | ✓ | ✓ | | 31.79 | 0.9361 | 0.0704 | 118.99 | 34.75 | 0.0559 | 0.0112 |
| Ours | ✓ | ✓ | ✓ | 32.04 | 0.9376 | 0.0684 | 118.99 | 34.75 | 0.0560 | 0.0112 |

**Table 6: Model complexity comparisons with previous model.**

| Method | DRSformer[15] | MPRNet[85] | Restormer[84] | BasicVSR++[8] | ESTINet [87] | Ours |
|---|---|---|---|---|---|---|
| GFLOPs | 1101.89 | 706.19 | 704.95 | 1616.44 | 681.83 | **118.99** |
| Runtime(s/frame) | 0.0381 | 0.0367 | 0.0696 | 0.0511 | 0.0341 | **0.0112** |
| Parameters(M) | 33.63 | **3.64** | 26.10 | 6.22 | 22.96 | 34.75 |

better temporal modeling ability, when compared to the optical flow, deformable kernel, and self-attention of SOTA compared methods.

*4.3.2 Quantitative Comparison on RainVID&SS Dataset.* Tab. 2 reports the quantitative results of our proposed method and 6 SOTA video deraining methods on the two testing sets, which are ImageNet-VID+ dataset and the Cam-Vid+ dataset. These 6 SOTA video deraining methods are MSCSC [35], FastDrain [30], PReNet[48], SLDNet[74], S2VD [83], and MPEVNet [54]. Regarding the ImageNet-VID+ dataset, our method achieves the largest PSNR score of 35.07 dB and the largest SSIM score of 0.9561. As the first place, our method outperforms the second place (i.e., MPEVNet) by a PSNR margin of 1.24 dB and a SSIM margin of 0.0109. On the Cam-Vid+ dataset, our video deraining network achieves the largest PSNR score of 32.65 dB and the largest SSIM score of 0.9328, which outperforms all compared six methods. It indicates that our method can more effectively remove rain streaks and has a better capability in significantly improving the image quality of input rainy videos with rain streaks.

*4.3.3 Quantitative Comparison on RainSynAll100 Dataset.* Tab. 3 reports the quantitative results of our network with 6 SOTA video deraining methods, and they are FastDerain [30], FCRVD [72], RMFD [73], BasicVSR++[8], NCFL [27], and SALN [79]. SALN has the better PSNR and SSIM scores among the 6 SOTA methods. Compared to SALN, our method improves the PSNR score from 29.78 dB to 32.16 dB, and enhances the SSIM score from 0.9315 to 0.9446. Our

superior PSNR and SSIM performance experimental results demonstrate our proposed technique significantly surpasses all competing methods in terms of removing rain streaks from rainy videos.

*4.3.4 Quantitative Comparison on LWDDS Dataset.* Tab. 4 summarizes the PSNR and SSIM scores of our proposed method and 6 SOTA video raindrops removal methods on the LWDDS datasets. These 6 SOTA video raindrops removal methods are CCN [47], Vid2Vid [61], VWR[65], BasicVSR++[8], ViMP-Net [68], and SALN [79]. From these quantitative results in Tab. 4, we can find that SALN has the largest PSNR score of 36.57 dB, and the largest SSIM score of 0.9802 among all compared 6 SOTA video raindrops removal methods. More importantly, our method further improves the PSNR score from 36.57 dB to 37.21 dB, and the SSIM score from 0.9802 to 0.9816. It also verifies the superior effect of our method in terms of the video raindrop removal task.

*4.3.5 Visual Comparisons.* Fig. 4 visually compares the results of removing rain streaks and raindrops from video frames on the VRDS dataset with different rainfall intensity and lighting conditions. Our network demonstrates superior performance in recovering clean background images, particularly excelling in raindrop areas where it more effectively restores detailed texture. These visual results demonstrate that our network excels in recovering image areas affected by raindrops and rain streaks. For instance, in the third sample set, our approach effectively maintains the coherence and completeness of the window area impacted by rain, while also retaining its natural coloration. Our network demonstrates superior preservation of non-rain background details by leveraging state space models for causally temporal modeling, combined with enhanced locality learning at both the sequence and patch levels. Moreover, Figure 5 shows the visual results of our network and state-of-the-art methods on real-world rainy video frames. These visual results indicate that our method can more effectively remove rain streaks and raindrops in real-world driving scenes and during outdoor video recordings, making it more applicable to common outdoor multimedia applications.

*4.3.6 Model Complexity and Efficiency Comparison.* As reported in Table 6, we compare the number of parameters, FLOPs, and running time of our network and state-of-the-art methods on a NVIDIA RTX 4090 GPU. The GFLOPs and Runtime are calculated by inferring a video clip of five frames with a resolution of 256×256. We follow [34] to calculate the GFLOPs metric. And the runtime indicates the time needed to process each frame during inference. Thanks to the linear complexity of SSMs and the critical components of our network, our approach achieves significant improvements in both effectiveness

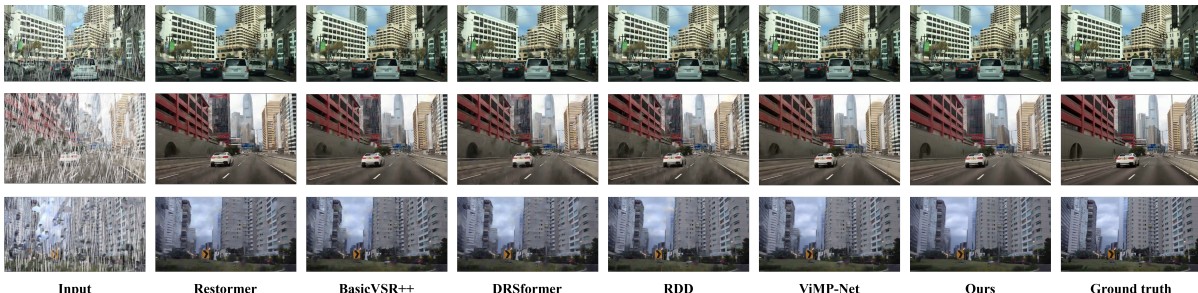

| Input | Restormer | BasicVSR++ | DRSformer | RDD | ViMP-Net | Ours | Ground truth |

**Figure 4: Visual comparisons of derained results from our network and state-of-the-art deraining methods on input video frames from the VRDS dataset. (Please zoom in for a better illustration.)**

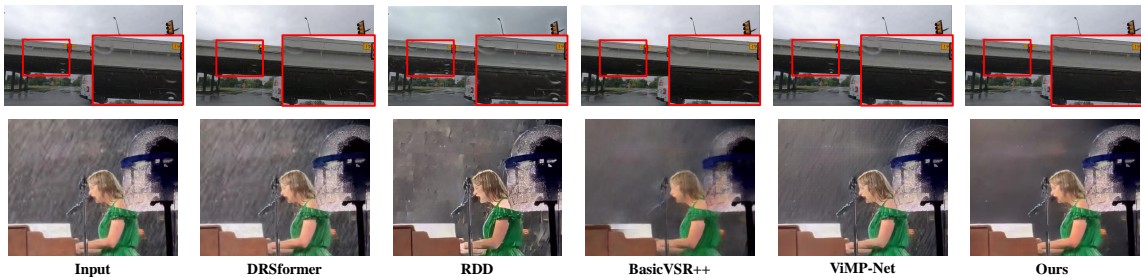

| Input | DRSformer | RDD | BasicVSR++ | ViMP-Net | Ours |

**Figure 5: Visual comparisons of derained results produced by our network and state-of-the-art deraining methods on input video frames from real-world rainy videos. (Please zoom in for a better illustration.)**

and efficiency. For more discussions on the model efficiency, please refer to *Supplementary Material* or our arXiv version.

### 4.4 Ablation Study

*4.4.1 Baseline Design.* We perform ablation experiments to verify the effectiveness of three critical components of our network, and they are the Global Mamba Block (GMB), the Local Mamba Block (LMB), and the Difference-Guided Dynamic Contrastive Locality Learning (DCL) of our RainMamba. Initially, we establish a baseline (denoted as "M1") by removing all three major components from our network. Subsequently, the Global Mamba Block is incorporated into the baseline model "M1" to build "M2". Following this, "M3" is constructed by integrating the Local Mamba Block into "M1". To build "M4", the Global Mamba Block is integrated into "M3", thereby resulting in a complete Coarse-to-Fine Mamba Module. Finally, we add our contrastive learning strategy to "M4" to reach a full setting of our RainMamba model.

*4.4.2 Quantitative Comparison.* Tab. 5 presents the quantitative results of our proposed method alongside the four baseline networks (i.e., "M1" through "M4") on the VRDS datasets. Specifically, compared with "M1", "M2" improves the PSNR score from 29.35 dB to 30.86 dB, the SSIM score from 0.9118 to 0.9277, and the LPIPS score from 0.0893 to 0.0774. It demonstrates the effectiveness of our Global Mamba Block in employing state space models to capture long-range dependencies among sequential frames. Also, "M3" demonstrates a further metric improvement, which indicates that the Local Mamba Block can improve the video restoration quality of our network by enhancing locality learning. Furthermore, "M4" significantly advances beyond M3, which proves the effectiveness of incorporating local scanning and Hilbert scanning mechanism together to model temporal information in a sequence-level manner. Moreover, our network further outperforms "M4", which indicates

**Table 7: Analysis of long video processing by our network on the NTURain dataset [11].**

| | 7 | 10 | 20 | 30 | 40 | 50 | 60 | 70 | 80 | 90 | 100 | 110 |
|---|---|---|---|---|---|---|---|---|---|---|---|---|
| PSNR | 37.534 | 37.722 | 37.806 | 37.838 | 37.847 | 37.858 | 37.863 | 37.867 | 37.870 | 37.875 | 37.876 | 37.875 |
| SSIM | 0.96904 | 0.97331 | 0.97357 | 0.97367 | 0.97371 | 0.97374 | 0.97376 | 0.97378 | 0.97379 | 0.97380 | 0.97380 | 0.97380 |
| Memory | 5,082M | 6,026M | 9,530M | 13,054M | 17,296M | 20,056M | 23,556M | 27,068M | 32,056M | 34,090M | 37,594M | 41,100M |

that leveraging our contrastive learning strategy benefits the locality learning of our network in a patch-level manner, thereby improving the video deraining performance of our network.

*4.4.3 Analysis of Long Video Processing.* We also explored the potential of our model in handling ultra long videos on NTURain dataset [11]. As reported in Table 7, we input full resolution video clips and compare the experimental results from using video clips of different lengths. This demonstrates that the long-sequence modeling capability of SSMs can effectively leverage the spatio-temporal contextual information in videos to successfully remove rain streaks. For more visual results and discussions about our experiments, please refer to *Supplementary Material* or our arXiv version.

## 5 CONCLUSION

In this work, we present a novel video deraining framework Rain-Mamba with the improved state space models. To the best of our knowledge, we are the first to apply state space models to achieve effective rain streaks and raindrops removal in videos. To better adapt Mamba to video deraining tasks, we introduce a Hilbert scanning mechanism to preserve the regional details based on the sequence level, which is the core element of our Local Mamba Block. Moreover, a difference-guided dynamic contrastive locality learning is designed to further enhance the local semantics from the patch level. Experimental results on four video-deraining benchmarking datasets demonstrate the superiority of our proposed framework. We believe this will be a compelling baseline for the future of state space models in the community of low-level vision tasks.

## ACKNOWLEDGMENTS

This work was supported by the Guangzhou-HKUST(GZ) Joint Funding Program (No. 2023A03J0671), This work is supported/funded by the Guangzhou Municipal Science and Technology Project (No. 2024A04J4230), the Guangzhou Industrial Information and Intelligent Key Laboratory Project (No. 2024A03J0628), the Nansha Key Area Science and Technology Project (No. 2023ZD003), and Guangzhou-HKUST(GZ) Joint Funding Program (No. 2024A03J0618).

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
