# OpenReview forum: "RainMamba: Enhanced Locality Learning with State Space Models for Video Deraining"
_acmmm.org/ACMMM/2024/Conference — MM2024 Oral_

### Official Review · Reviewer_n3eQ · 2024-04-30

**Rating:** 4
**Confidence:** 2

**Summary:**

This paper proposes an improved state-space models-based video deraining network called RainMamba with a Hilbert scanning mechanism to capture sequence-level local information. In addition, a difference-guided dynamic contrastive locality learning strategy is introduced to enhance the patch-level self-similarity learning ability. Experimental results on synthesized and real-world videos demonstrate the superiority of the proposed method.

**Strengths:**

1. The main methodology is easy to read and understand.
2. This paper introduces a Hilbert scanning mechanism to enhance locality learning for Mamba-based video deraining task.
3. Experimental results on synthesized and real-world datasets prove the effectiveness of the proposed method compared to previous approaches.

**Limitations:**

1. Relevant references for Hilbert scanning mechanism require to be included.

2. The parameters of different methods in Table 5 should be added for a more comprehensive comparison.

3. It appears that the proposed method did not investigate the impact of different Hilbert-based scanning directions on model performance. Additionally, there are alternative scanning methods available, so it's worth explaining why the zigzag order approach was chosen for the global Mamba block.

**Suitability:**

2

---

### Official Review · Reviewer_Gke2 · 2024-05-15

**Rating:** 5
**Confidence:** 4

**Summary:**

1. Hilbert scanning mechanism to capture sequence-level local information.
2. Difference-guided dynamic contrastive locality learning to enhance patch-level self-similarity learning.
3. Experiments across synthesized and real-world video deraining datasets validate the superiority of the proposed method.

**Strengths:**

1. A novel scanning mechanism to help Mamba enhance local relationships, which is a crucial prior for vision tasks.
2. New SoTA performance across synthesized and real-world deraining datasets.
3. The paper is well-organized and easy to follow.

**Limitations:**

1. Can you provide more theoretical interpretation about local prior implied in Hilbert curve? One point of this paper is use the local prior of Hilbert scanning. Therefore, I think more solid evidence will convince readers.
2. Modelling local relationships is not a challenging for existing visual backbones, such as CNN can inherently capture local correlations. Hence, I think that the motivation of designing complex scanning rule to just capture local relationships is weak.
3. I suggest the authors add efficiency comparison with previous methods.

Given the well-presentation, new scanning method and solid results of this paper, I lean towards acceptance, but encourage the authors to consider these limitations carefully.

**Suitability:**

2

---

### Official Review · Reviewer_FrMR · 2024-05-23

**Rating:** 5
**Confidence:** 4

**Summary:**

This paper first presents a new approach for video deraining based on the state space models. The authors introduce a novel Hilbert scanning mechanism that effectively addresses the pixel forgetting issue found in traditional Mamba-based methods by minimizing the distance between spatio-temporally adjacent pixels during their conversion to 1D sequences. Additionally, the paper proposes a difference-guided dynamic contrastive locality learning mechanism that enhances the model’s ability to perceive intra- and inter-frame spatial correlations. Experimental results validate the efficacy of the proposed method, indicating significant improvements in handling video deraining tasks.

**Strengths:**

1)	A new pipeline based on state space models for video deraining tasks is proposed.
2)	Based on the mathematical characterization of space-filling curves, the authors propose a new scanning approach to enhance the local learning capability of the network. It’s interesting to introduce 3d Hilbert scan into the mamba-based network.
3)	Moreover, the difference-guided dynamic contrastive locality learning mechanism can finetune the network based on the non-local prior.
4)	The experimental results show that results on four synthetic video deraining benchmarks and real-world data, surpassing previous works.

**Limitations:**

1)	It would be helpful if the authors could provide more details about the motivation behind the setting of the dynamic learning in the proposed contrastive learning mechanism.
2)	Could the authors explain the more definition of local and global scanning mechanisms. What is difference between the proposed local scanning and that in LocalMamba.
3)	Minor: There are some typos in this article. For example, In line 273, there is an unnecessary repetition of the word "the." Please further refine.
4)	The appendix indicates that your method may outperform others in terms of efficiency, as evidenced by its ranking in Table 1 of the Appendix. Could the authors provide more details on the factors contributing to this efficiency?
5)	In Table 5, the author should describe the meaning of GMB, LMB, DCL in the caption.

**Suitability:**

3

---

### Official Review · Reviewer_Waij · 2024-05-27

**Rating:** 6
**Confidence:** 4

**Summary:**

The authors present an improved SSMs- based video deraining network (RainMamba) with a novel Hilbert scanning mechanism to better capture sequence-level local infor mation. They also introduce a difference-guided dynamic contrastive locality learning strategy to enhance the patch-level self-similarity learning ability of the proposed network. Extensive experiments on four synthesized video deraining datasets and real-world rainy videos demonstrate the superiority of our network in the removal of rain streaks and raindrops.

**Strengths:**

1.  Significant experimental results: Extensive experiments on four synthetic video rain removal datasets and real-world rainy day videos show that the proposed network outperforms the existing state-of-the-art methods in removing rain streaks and raindrops, with significant performance improvement.
2.  Methodological innovation: The introduction of Hilbert scanning mechanism and difference-guided dynamic contrast local learning strategy solves the problem of local information loss in traditional SSM for video processing in a novel and effective way.
3. Combination of theory and practice: this paper not only proposes new models and mechanisms theoretically, but also verifies their effectiveness in practical applications through a large number of experiments, which has high academic and application value.

**Limitations:**

Despite the introduction of the Hilbert scanning mechanism and the dynamic comparison learning strategy, the high implementation complexity of these methods may increase the training and inference time of the model, which affects the feasibility of practical applications.

**Suitability:**

3

---

### Meta-Review · Area_Chair_MEzt · 2024-07-03

**Recommendation:** Accept (Oral)
**Confidence:** 5

**Metareview:**

All reviewers concur that the submission should be accepted and are confident in their review, so I see no reason not to accept this paper.